# MAE: Mutual Posterior-Divergence Regularization for Variational AutoEncoders

**Xuezhe Ma, Chunting Zhou & Eduard Hovy**
Carnegie Mellon University
`{xuezhem, ctzhou, ehovy}@cs.cmu.edu`

## Abstract

Variational Autoencoder (VAE), a simple and effective deep generative model, has led to a number of impressive empirical successes and spawned many advanced variants and theoretical investigations. However, recent studies demonstrate that, when equipped with expressive generative distributions (aka. decoders), VAE suffers from learning uninformative latent representations with the observation called *KL Varnishing*, in which case VAE collapses into an unconditional generative model. In this work, we introduce *mutual posterior-divergence regularization*, a novel regularization that is able to control the geometry of the latent space to accomplish meaningful representation learning, while achieving comparable or superior capability of density estimation. Experiments on three image benchmark datasets demonstrate that, when equipped with powerful decoders, our model performs well both on density estimation and representation learning.

## 1 Introduction

Representation learning, besides data distributions estimation, is a principle component in generative models. The goal is to identify and disentangle the underlying causal factors, to tease apart the underlying dependencies of the data, so that it becomes easier to understand, to classify, or to perform other tasks (Bengio et al., 2013). Among these generative models, VAE (Kingma & Welling, 2014; Rezende et al., 2014) gains popularity for its capability of estimating densities of complex distributions, while automatically learning meaningful (low-dimensional) representations from raw data. VAE, as a member of latent variable models (LVMs), defines the joint distribution between the observed data (visible variables) and a set of latent variables by factorizing it as the product of a prior over the latent variables and a conditional distribution of the visible variables given the latent ones (detailed in §2). VAEs are usually estimated by maximizing the likelihood of the observed data by marginalizing over the latent variables, typically via optimizing the evidence lower bound (ELBO). By learning a VAE from the data with the appropriate hierarchical structure of latent variables, the hope is to uncover and untangle the causal sources of variations that we are interested in.

A notorious problem of VAEs, however, is that the marginal likelihood may not guide the model to learn the intended latent variables. It may instead focus on explaining irrelevant but common correlations in the data (Ganchev et al., 2010). Extensive previous studies (Bowman et al., 2015; Chen et al., 2017a; Yang et al., 2017) showed that optimizing the ELBO objective is often completely disconnected from the goal of learning good representations. An extreme case called *KL varnishing*, happens when using sufficiently expressive decoding distributions such as auto-regressive ones; the latent variables are often completely ignored and the VAE regresses to a standard auto-regressive model (Larochelle & Murray, 2011; Oord et al., 2016).

This problem has spawned significant interests in analyzing and solving it from both theoretical and practical perspectives. We can only name a few here due to space limits. Some previous work (Bowman et al., 2015; Sønderby et al., 2016b; Serban et al., 2017) attributed the KL varnishing phenomenon to "optimization challenges" of VAEs, and proposed training methods including annealing the relative weight of the KL term in ELBO (Bowman et al., 2015; Sønderby et al., 2016b) or adding *free bits* (Kingma et al., 2016; Chen et al., 2017a). However, Chen et al. (2017a) pointed out that this phenomenon arises not just due to the optimization challenges, and even if we find the exact solution for the optimization problems, the latent code will still be ignored at optimum. They

proposed a solution by limiting the capacity of the decoder and applied PixelCNN (Oord et al., 2016) with small local receptive fields as the decoder of VAEs to model 2D images, achieving both impressive performance for density estimation and informative latent representations. Yang et al. (2017) embraced a similar idea and applied VAE to text modeling by using dilated CNN as the decoder. Unfortunately, these approaches require manual and problem-specific design of the decoder's architecture to learn meaningful representations. Other studies attempted to explore alternatives of ELBO. Makhzani et al. (2015) proposed Adversarial Autoencoders (AAEs) by replacing the KL-divergence between the posterior and prior distributions with Jensen-Shannon divergence on the aggregated posterior distribution. InfoVAEs (Zhao et al., 2017) generalized the Jensen-Shannon divergence in AAEs to a divergence family and linked its objective to the Mutual Information between the data and the latent variables. However, directly optimizing these objectives is intractable, requiring advanced approximate learning methods such as adversarial learning or Maximum-Mean Discrepancy (Gretton et al., 2007; Dziugaite et al., 2015; Li et al., 2015). Moreover, these models' performance on density estimation significantly falls behind state-of-the-art models (Salimans et al., 2017; Chen et al., 2017a).

In this paper, we propose to tackle the aforementioned representation learning challenges of VAEs by adding a data-dependent regularization to the ELBO objective. Our contributions are three-fold: (1) Algorithmically, we introduce the *mutual posterior-divergence regularization* for VAEs, named MAEs (§3.2), to control the geometry of the latent space during learning by encouraging the learned variational posteriors to be diverse (i.e. they are favored to be mutually "different" from each other), to achieve low-redundant, interpretable representation learning. (2) Theoretically, we establish a close relation between MAE and InfoVAE, by showing that the mutual posterior-devergence regularization maximizes a symmetric version of the KL divergence involved in InfoVAE's mutual information term (§3.3). (3) Experimentally, on three benchmark datasets for images, we demonstrate the effectiveness of MAE as a density estimator by state-of-the-art log-likelihood results on MNIST and OMNIGLOT, and comparable result on CIFAR-10. Moreover, by performing image reconstruction, unsupervised and semi-supervised classification, we show that MAE is also capable of learning meaningful latent representations, even combined with a sufficiently powerful decoder (§4).

## 2 VARIATIONAL AUTOENCODERS

### 2.1 NOTATIONS

Throughout we use uppercase letters for random variables, and lowercase letters for realizations of the corresponding random variables. Let $X \in \mathcal{X}$ be the randoms variables of the observed data, e.g., $X$ is an image or a sentence for image and text generation, respectively.

Let $P$ denote the true distribution of the data, i.e., $X \sim P$, and $D = \{x_1, \ldots, x_N\}$ be our training sample, where $x_i, i = 1, \ldots, N$, are usually i.i.d. samples of $X$. Let $\mathcal{P} = \{P_\theta : \theta \in \Theta\}$ denote a parametric statistical model indexed by parameter $\theta \in \Theta$, where $\Theta$ is the parameter space. $p$ is used to denote the density of corresponding distribution $P$. In the literature of deep generative models, deep neural networks are the most widely used parametric models. The goal of generative models is to learn the parameter $\theta$ such that $P_\theta$ can best approximate the true distribution $P$.

### 2.2 VAEs

In the framework of VAEs, or general LVMs, a set of latent variables $Z \in \mathcal{Z}$ are introduced to characterize the hidden patterns of $X$, and the model distribution $P_\theta(X)$ is defined as the marginal of the joint distribution between $X$ and $Z$:

$$p_\theta(x) = \int_{\mathcal{Z}} p_\theta(x, z) d\mu(z) = \int_{\mathcal{Z}} p_\theta(x|z) p_\theta(z) d\mu(z), \quad \forall x \in \mathcal{X}, \tag{1}$$

where the joint distribution $p_\theta(x, z)$ is factorized as the product of a prior $p_\theta(z)$ over the latent $Z$, and the "generative" distribution $p_\theta(x|z)$. $\mu(z)$ is the base measure on the latent space $\mathcal{Z}$. Typically, prior $p_\theta(z)$ is modeled with a simple distribution like multivariate Gaussian, or transforming simple priors to complex ones by normalizing flows and variants (Rezende & Mohamed, 2015; Kingma et al., 2016; Sønderby et al., 2016a).

To learn parameters $\theta$, we wish to minimizes the negative log-likelihood of the parameters:

$$\min_{\theta \in \Theta} \frac{1}{N} \sum_{i=1}^{N} -\log p_\theta(x_i) = \min_{\theta \in \Theta} \mathrm{E}_{\tilde{P}(X)}[-\log p_\theta(X)] \tag{2}$$

where $\tilde{P}(X)$ is the empirical distribution derived from training data $D$. In general, this marginal likelihood is intractable to compute or differentiate directly for high-dimensional latent space $\mathcal{Z}$. Variational Inference (Wainwright et al., 2008) provides a solution to optimize the *evidence lower bound* (ELBO) an alternative objective by introducing a parametric *inference model* $q_\phi(z|x)$:

$$\begin{aligned}
\mathcal{L}_{elbo}(\theta, \phi) &= \mathrm{E}_{p(X)} \left[ -\mathrm{E}_{q_\phi(Z|X)}[\log p_\theta(X|Z)] + \mathrm{KL}(q_\phi(Z|X)||p_\theta(Z)) \right] \\
&= \mathrm{E}_{p(X)} \left[ -\log p_\theta(X) + \mathrm{KL}(q_\phi(Z|X)||p_\theta(Z|X)) \right] \geq \mathrm{E}_{p(X)} \left[ -\log p_\theta(X) \right]
\end{aligned} \tag{3}$$

where $\mathcal{L}_{elbo}$ could be seen as an autoencoding loss with $q_\phi(z|x)$ being the encoder and $p_\theta(x|z)$ being the decoder, with the first term in the RHS in (3) as the reconstruction error.

## 2.3 AUTOENCODING PROBLEM IN VAES

As discussed in Chen et al. (2017a), without further assumptions, the ELBO objective $\mathcal{L}_{elbo}$ in (3) may not guide the model towards the intended role for the latent variables $Z$, or even learn uninformative $Z$ with the observation that the KL term $\mathrm{KL}(q_\phi(Z|X)||p_\theta(Z))$ varnishes to zero. For example, suppose we use an auto-regressive decoder, $p_\theta(x|z) = \prod_i p_\theta(x_i|x_{<i}, z)$, which is sufficiently expressive that it can model the data distribution $P(X)$ without the assistance of $Z$, i.e, $p_\theta(x_i|x_{<i}, z) = p_\theta(x_i|x_{<i})$. In this case, the optimal $Z$ w.r.t. $\mathcal{L}_{elbo}$ is the one independent with $X$, with the inference model reducing to the prior, i.e., $q_\phi(Z|X) = p_\theta(Z), \forall X \in \mathcal{X}$.

The essential reason of this problem is that, under absolutely unsupervised setting, the marginal likelihood based objective $\mathcal{L}_{elbo}$ incorporates no (direct) supervision on the latent space to characterize the latent variable $Z$ with preferred properties w.r.t. representation learning. The main goal of this work is to explicitly control the geometry of the latent space, in the hope that preferred latent representations would be characterized and selected.

# 3 MUTUAL POSTERIOR-DIVERGENCE REGULARIZATION

## 3.1 GEOMETRIC PROPERTIES OF MEANINGFUL LATENT SPACE

Motivated by the Diversity-Inducing Mutual Angular Regularization (Xie et al., 2015) which is widely used in LVMs, we propose to regularize the posteriors $q_\phi(z|x)$ of different data $x \in \mathcal{X}$, to encourage them to diversely, smoothly, and evenly spread out in the data space $\mathcal{X}$. The intuition is: (1) To make posteriors mutually diverse from each other, the patterns captured by different posteriors are likely to have less redundancy and hence characterizing and interpreting different data $x$. (2) To make posteriors smoothly and evenly distributed in the whole space of $\mathcal{X}$, the shared patterns of similar data points are likely to be captured by their posteriors, to avoid isolating each data point from others. By balancing the diversity and smoothness of the distribution of posteriors, learned representations are encouraged to maintain global structured information, discarding detailed texture of local dependencies in the data.

## 3.2 MAES

**Measure of Diversity.** We propose to use expectation of the mutual KL-divergence between a pair of data to measure the diversity of posteriors. Specifically, the mutual posterior diversity is defined as:

$$MPD = \mathrm{E}_{X_1, X_2 \sim P(X)}[\mathrm{KL}(q_\phi(Z|X_1)||q_\phi(Z|X_2))] \tag{4}$$

There are two main reasons we use KL-divergence instead of others as the measure of diversity: (1) KL-divergence is transformation invariant, i.e., for an invertible smooth function $f$,

$$\mathrm{KL}(q_\phi(Z|X_1)||q_\phi(Z|X_2)) = \mathrm{KL}(q_\phi(f(Z)|X_1)||q_\phi(f(Z)|X_2))$$

It makes the computation efficient for complex posteriors that are transformed from simple ones, such as applying normalizing flows and variants (Rezende & Mohamed, 2015; Kingma et al., 2016;

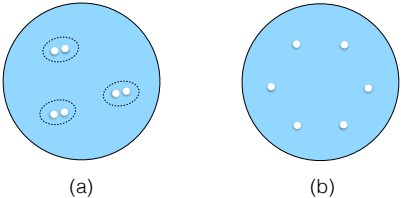

Figure 1: The mean of pairwise distances between points in (a) is close to (b), while the standard deviation in (a) is much larger.

Sønderby et al., 2016a). (2) KL-divergence has a close relation with mutual information, an important information-theoretic measure of the mutual dependence between two variables, which provides us the theoretical justification of the proposed regularizer (detailed in §3.3).

In MAEs, we propose to maximize the mutual posterior diversity (MPD) in (4). A straightforward way is to add negative MPD to the objective $\mathcal{L}_{elbo}$ in (3) that VAEs attempt to minimize. There are, however, two practical issues: (1) The scale of MPD, particularly for continuous $Z$, is much larger than that of $\mathcal{L}_{elbo}$. We need to choose a hyperparameter carefully to control the scale of MPD, making optimization much more challenging and unstable. (2) For multivariate $Z$, e.g. a $K$-dimensional $Z = (Z_1, Z_2, \ldots, Z_K)$, due to the property of KL-divergence, MPD may be dominated by a small group of dimensions, leaving others close to zero. In this case, most dimensions of $Z$ are uninformative, which is not a desired representation.

To solve the two problems, in practice, we propose to minimize a MPD-based loss instead of directly maximizing MPD itself:

$$\mathcal{L}_{diverse} = \mathrm{E}_{X_1, X_2 \sim P(X)} \left[ \sum_{k=1}^{K} \log(1 + \exp(-\mathrm{KL}(q_\phi(Z_k|X_1)||q_\phi(Z_k|X_2)))) \right] \qquad (5)$$

$\mathcal{L}_{diverse}$ has two important good properties: (1) $\mathcal{L}_{diverse} \geq 0$. (2) $\mathcal{L}_{diverse} \to 0$ iff $\mathrm{KL}(q_\phi(Z_k|X_1)||q_\phi(Z_k|X_2)) \to \infty, \forall k$. The first property sets a lower bound of $\mathcal{L}_{diverse}$, making optimization much more stable. The second one guarantees that all the dimensions of the latent $S$ need to be mutually diverse w.r.t minimizing $\mathcal{L}_{diverse}$.

**Measure of Smoothness.** The smoothness of the distribution of posteriors is measured by utilizing the standard deviation of the mutual KL-divergence:

$$\mathcal{L}_{smooth} = \mathrm{STD}_{X_1, X_2 \sim P(X)}[\mathrm{KL}(q_\phi(Z|X_1)||q_\phi(Z|X_2))] \qquad (6)$$

where STD stands for standard deviation of random variables.

$\mathcal{L}_{smooth}$ encourages the posteriors to smoothly and evenly spread out to different directions. Encouraging the standard deviation to be small can prevent the phenomenon that the posteriors fall into several small groups that are isolated from each other. It is crucially important for unsupervised clustering tasks, in which we want to cluster similar data into a big group instead of splitting them into multiple separated small groups (see §4.1.2 for detailed experimental results). Figure 1 shows two sets of distributions of data points, where the mean of the pairwise distances of the first set (Figure 1 (a)) is roughly the same as the second set (Figure 1 (b)). But the standard deviation of the first set is larger.

In the framework of MAEs, the final objective to minimize is:

$$\mathcal{L}_{MAE} = \mathcal{L}_{elbo} + \eta \mathcal{L}_{diverse} + \gamma \mathcal{L}_{smooth} \qquad (7)$$

where $\eta > 0, \gamma > 0$ are regularization constants to balance the three losses in $\mathcal{L}_{MAE}$. Even though MAE introduces two extra hyperparameters $\eta$ and $\gamma$, we find them easy to tune and MAE shows robust performance with different values of $\eta$ and $\gamma$.

To solve (7), we can approximate $\mathcal{L}_{diverse}$ and $\mathcal{L}_{smooth}$ using Monte carlo in each mini-batch:

$$\mathcal{L}_{diverse} \approx \frac{1}{M} \sum_{x_1 \neq x_2} \sum_{k=1}^{K} \log(1 + \exp(-\mathrm{KL}(q_\phi(Z_k|x_1)||q_\phi(Z_k|x_2)))) \qquad (8)$$

where $M$ is the number of valid pairs of data in each mini-batch. $\mathcal{L}_{smooth}$ is approximately computed similarly.

### 3.3 THEORETICAL JUSTIFICATION

So far our discussion has been concentrated on the motivation and mathematical formulation of the proposed regularization method for VAE. In this section, we provide theoretical justification by connecting the mutual posterior diversity (MPD) in (4) with the mutual information term defined in InfoVAE (Zhao et al., 2017). With the end goal of theoretically justifying the proposed regularizer in mind, we first review the background of the mutual information (MI) term involved in the InfoVAE objective, which is central for linking MAE and InfoVAE.

**Mutual Information Maximization.** InfoVAE proposed the mutual information by first defining the joint "inference distribution":

$$q_\phi(x, z) = p(x)q_\phi(z|x)$$

where $p(x)$ is the density of the true data distribution $P(X)$. Then they added a mutual information maximization term that prefers high mutual information between $X$ and $Z$ under $q_\phi(x, z)$ to the standard $\mathcal{L}_{elbo}$:

$$\mathcal{L}_{InfoVAE} = \mathcal{L}_{elbo} - \lambda I_{q_\phi(x,z)}(x; z)$$

and further proved that

$$I_{q_\phi(x,z)}(x; z) = \mathrm{E}_{P(X)}[\mathrm{KL}(q_\phi(z|x)||q_\phi(z))] \tag{9}$$

where $q_\phi(z) = \int_{\mathcal{X}} q_\phi(x, z)d\mu(x)$ is the marginal of $q_\phi(x, z)$. Mutual information inspired objectives have been explored in GANs (Goodfellow et al., 2014; Chen et al., 2016), clustering (Hinton et al., 1995; Krause et al., 2010) and representation learning (Esmaeili et al., 2018; Hjelm et al., 2018).

**Relation between MPD and MI.** The following theorem states our major result that reveals the relation between MPD and MI (proof in Appendix A):

**Theorem 1.** *The mutual posterior diversity (MPD) in* (4) *is a symmetric version of the KL-divergence of MI in* (9):

$$MPD = \mathrm{E}_{P(X)}[\mathrm{KL}(q_\phi(z|x)||q_\phi(z)) + \mathrm{KL}(q_\phi(z)||q_\phi(z|x))] \tag{10}$$

Roughly, Theorem 1 states that maximizing MPD and MI achieve the same goal: maximizing the divergence between the posterior distribution $q_\phi(z|x)$ and the marginal $q_\phi(z)$. Note that the (approximate) computation of MPD, as described in (8), is much easier than MI, which is generally intractable and requires adversarial learning or Maximum-Mean Discrepancy.

## 4 EXPERIMENTS

In this paper, we choose Variational Lossy Autoencoder (VLAE) (Chen et al., 2017a), VAE with auto-regressive flow (AF) prior, and auto-regressive decoder, as the basic architecture of our MAE models. More detailed descriptions, results, and analysis of the conducted experiments are provided in Appendix B.

### 4.1 BINARY IMAGES

We evaluate MAE on two binary images that are commonly used for evaluating deep generative models: MNIST (LeCun et al., 1998) and OMNIGLOT (Lake et al., 2013; Burda et al., 2015), both with dynamically binarized version (Burda et al., 2015). VLAE networks used in binary image datasets are similar of that described in Chen et al. (2017a): ResNet (He et al., 2016) encoder same as in ResNet VAE (Kingma et al., 2016), PixelCNN (Oord et al., 2016) decoder with 6 layers of masked convolution, and 32-dimensional latent code with AF prior implemented with MADE (Germain et al., 2015). The only difference is that the PixelCNN decoder has varying filter sizes: two 7x7 layers, followed by two 5x5 layers, and finally two 3x3 layers, instead of a fixed filter size of 3x3 used in Chen et al. (2017a). Hence the decoder we use has larger receptive field, to ensure that the decoder is sufficiently expressive. The same architecture is applied to all the experiments on both the two datasets. For pair comparison, we re-implemented VLAE using the same architecture in our MAE model. "Free bits" (Kingma et al., 2016) is used to improve optimization stability of VLAE (not for MAE). For hyperparameters $\eta$ and $\gamma$, we explored a few configurations: $\eta$ is selected from $[0.5, 1.0, 2.0]$, and $\gamma$ from $[0.1, 0.5, 1.0]$.

Table 1: Image modeling results on dynamically binarized MNIST and OMNIGLOT.

(a) MNIST

| Model | NLL(KL) |
|---|---|
| IWAE (Burda et al., 2015) | 82.90 |
| LVAE (Sønderby et al., 2016a) | 81.74 |
| InfoVAE (Zhao et al., 2017) | 80.76 |
| Discrete VAE (Rolfe, 2016) | 80.04 |
| IAF VAE (Kingma et al., 2016) | 79.10 |
| VLAE (Chen et al., 2017a) | 78.53 |
| VLAE (re-impl) | 78.26 (9.02) |
| MAE: $\eta = 1.0, \gamma = 0.1$ | 78.02 (11.38) |
| MAE: $\eta = 0.5, \gamma = 0.5$ | 78.00 (10.44) |
| MAE: $\eta = 1.0, \gamma = 0.5$ | **77.98 (11.54)** |
| MAE: $\eta = 2.0, \gamma = 0.5$ | 77.99 (12.67) |
| MAE: $\eta = 1.0, \gamma = 1.0$ | 78.15 (10.19) |

(b) OMNIGLOT

| Model | NLL(KL) |
|---|---|
| IWAE (Burda et al., 2015) | 103.38 |
| LVAE (Sønderby et al., 2016a) | 102.11 |
| Discrete VAE (Rolfe, 2016) | 97.43 |
| SA-VAE (Kim et al., 2018) | 90.05 (2.78) |
| VLAE (Chen et al., 2017a) | 89.83 |
| VampPrior (Tomczak, 2018) | 89.76 |
| VLAE (re-impl) | 89.62 (8.43) |
| MAE: $\eta = 0.5, \gamma = 0.1$ | 89.21 (9.08) |
| MAE: $\eta = 0.5, \gamma = 0.2$ | **89.09 (12.66)** |
| MAE: $\eta = 1.0, \gamma = 0.2$ | 89.15 (14.86) |
| MAE: $\eta = 0.5, \gamma = 0.5$ | 89.41 (10.53) |

Table 2: Performance of unsupervised clustering and semi-supervised classification. For each experiment, we report the average over 5 runs.

| | unsupervised clustering | | | semi-supervised classification | | | | | |
| | K-Means | | | | KNN | | | Linear | |
| Model | K=10 | K=20 | K=30 | 100 | 1000 | All | 100 | 1000 | All |
|---|---|---|---|---|---|---|---|---|---|
| ResNet VAE w. AF | 67.3 | 81.6 | 86.6 | 77.4 | 94.3 | 98.1 | 84.6 | 94.3 | 97.4 |
| VLAE | 68.1 | 74.0 | 79.1 | 75.7 | 90.0 | 95.6 | 86.4 | 93.7 | 96.1 |
| MAE: $\eta = 1.0, \gamma = 0.1$ | 82.7 | 92.3 | 93.0 | 86.6 | 95.5 | 97.8 | 91.1 | 96.3 | 98.3 |
| MAE: $\eta = 0.5, \gamma = 0.5$ | 84.7 | **92.6** | 93.2 | 86.3 | 96.3 | 98.0 | 90.6 | 96.1 | 98.1 |
| MAE: $\eta = 1.0, \gamma = 0.5$ | **91.2** | **92.6** | 93.6 | **86.7** | 95.9 | **98.2** | **91.5** | **96.4** | **98.4** |
| MAE: $\eta = 2.0, \gamma = 0.5$ | 78.2 | 92.0 | 92.8 | 85.5 | 96.4 | **98.2** | 90.7 | 96.0 | 98.0 |
| MAE: $\eta = 2.0, \gamma = 1.0$ | 83.1 | 92.3 | **94.3** | 86.2 | **96.6** | 98.1 | 90.0 | 95.7 | 98.0 |

### 4.1.1 DENSITY ESTIMATION

We first evaluate MAE on density estimation performance. Table 1 provides the results of MAE with different settings of hyperparameters on MNIST, together with previous top systems for comparison. Reported marginal negative log-likelihood (NLL) is evaluated with 4096 importance samples (Burda et al., 2015). Our MAE achieves state-of-the-art performance on both the two datasets, exceeding all previous models and the re-implemented VLAE. Note that our re-implementation of VLAE obtains better performance than the original one in Chen et al. (2017a), demonstrating the effectiveness of increasing decoder expressiveness by enlarging its receptive field.

### 4.1.2 REPRESENTATION LEARNING

In order to evaluate the quality of the learned latent representations, we conduct three sets of experiments — image reconstruction and generation, unsupervised clustering, and semi-supervised classification.

**Image Reconstruction and Generation.** The visualization of the of image reconstruction and generation on MNIST and OMNIGLOT is shown in Figure 2 and Figure 3. For comparison, we also show the reconstructed images from VLAE. MAE achieves better reconstruction ability than VLAE, proving that the latent code from MAE encodes more information from data.

**Unsupervised Clustering.** As discussed above, good latent representations need to capture global structured information and disentangle the underlying causal factors, rather than just memorizing the data. From this perspective, good image reconstruction results cannot guarantee good representations. To further evaluate the quality of the learned representation from MAE, we conduct the experiments of unsupervised clustering on MNIST. We perform K-Means clustering algorithm (Hartigan & Wong, 1979) on the learned representations. The class label of each cluster is assigned by finding the closest

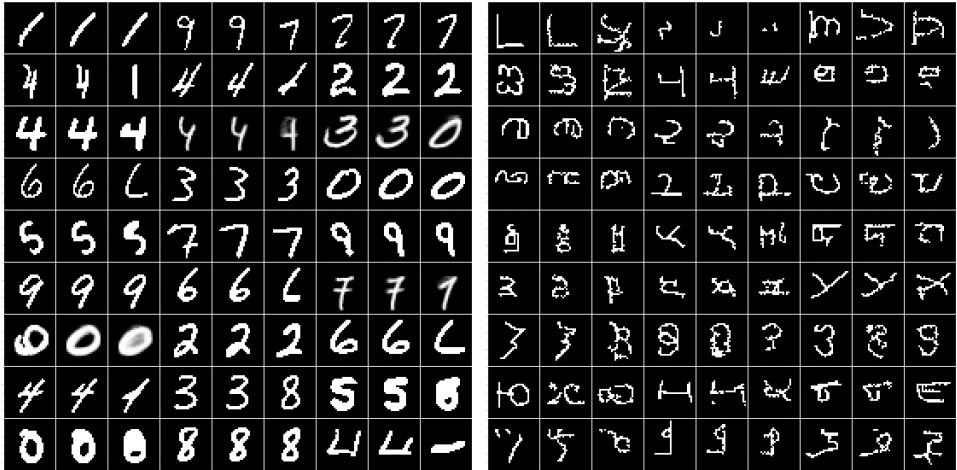

(a) MNIST reconstruction  (b) OMNIGLOT reconstruction

Figure 2: Image reconstructions on MNIST and OMNIGLOT. Every three columns compose a set of reconstruction, original image is on the left, reconstructed image from MAE is in the middle, and reconstructed one from VLAE is on the right.

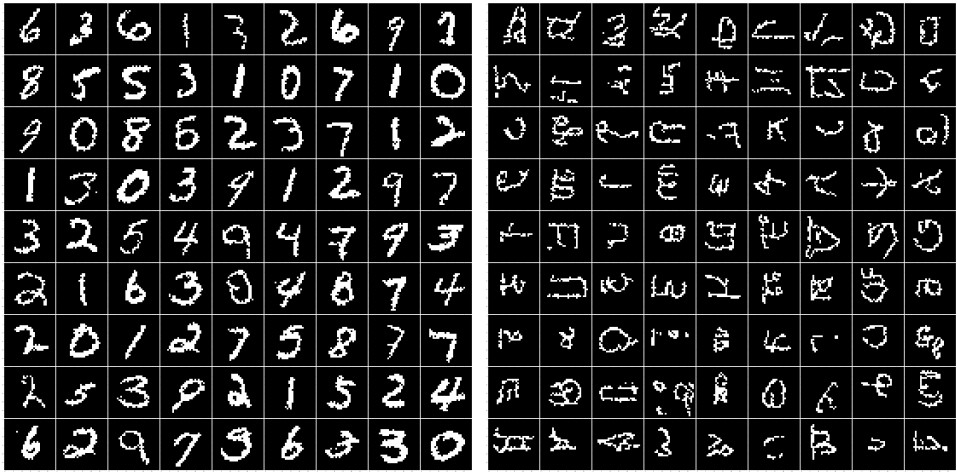

(a) MNIST samples from MAE  (b) OMNIGLOT samples from MAE

Figure 3: Image samples on MNIST and OMNIGLOT from MAE.

sample in the training data with the cluster head. Evaluation of clustering accuracy is based on the assigned cluster labels. We run three experiments with $K \in [10, 20, 30]$.

Table 2 (left section) illustrates the clustering performance. To make a thorough comparison, we also re-implemented a VAE model with a factorized decoder $p_\theta(x|z) = \prod_i p_\theta(x_i|z)$ and AF prior, which has been proven to obtain remarkable reconstruction performance. The VAE model uses ResNet (He et al., 2016) as its encoder and decoder similar to Chen et al. (2017a). From Table 2 we see that MAE significantly outperforms ResNet VAE and VLAE, especially when the number of clusters $K$ is small. Interestingly, when $K$ keeps increasing, clustering accuracy of ResNet VAE increases rapidly, showing that in its latent space the data are split into small groups.

In addition, towards the affects of $\eta$ and $\gamma$ on learned representations, MAEs with larger $\eta$ obtain worse performance on $K = 10$. The reason might be that large $\eta$ encourages the posteriors to diverse from each other, by splitting the data into small groups. Meanwhile, increasing $\gamma$ is effective to

Table 3: Density estimation performance on CIFAR-10. Negative log-likelihood is evaluated with 512 importance samples.

| Model | bits/dim |
|---|---|
| Deep GMMs (Van den Oord & Schrauwen, 2014) | 4.00 |
| Real NVP (Dinh et al., 2016) | 3.49 |
| PixelCNN (Oord et al., 2016) | 3.14 |
| PixelRNN (Oord et al., 2016) | 3.00 |
| PixelCNN++ (Salimans et al., 2017) | 2.92 |
| PixelSNAIL (Chen et al., 2017b) | **2.85** |
| Conv DRAW (Gregor et al., 2016) | 3.50 |
| IAF VAE (Kingma et al., 2016) | 3.11 |
| VLAE (Chen et al., 2017a) | 2.95 |
| VLAE (re-impl) | 2.98 |
| MAE: $\eta = 0.5, \gamma = 1.5$ | **2.95** |
| MAE: $\eta = 0.5, \gamma = 2.0$ | 2.97 |
| MAE: $\eta = 1.0, \gamma = 2.0$ | 2.96 |

prevent the phenomenon, showing that in practice considerations on the trade-off between space diversity and smoothness are needed.

**Semi-supervised Classification.** For semi-supervised classification, we re-implemented the M1 model as described in Kingma et al. (2014). To test quality of information encoded in the latent representations, we choose two simple classifiers with limited capacity — K-nearest neighbor ($K = 10$) and linear logistic regression. For each classifier, we use different numbers of labeled data — 100, 1000 and all the training data from MNIST.

From the results listed in Table 2 (right section), MAE obtains the best classification accuracy on all the settings. Moreover, the improvements of MAE over ResNet VAE and VLAE are more significant when the number of labeled training data is small, further proving the meaningful representation learned from MAE.

## 4.2 NATURAL IMAGES

In addition to binary image datasets, we also applied MAE to CIFAR-10 dataset (Krizhevsky & Hinton, 2009) of natural images. The VLAE with DenseNet (Huang et al., 2017) encoder and PixelCNN++ (Salimans et al., 2017) decoder described in Chen et al. (2017a) is used as the neural architecture of MAE. To ensure that the decoder is sufficiently expressive, the decoder PixelCNN has 5 blocks of 96 feature maps and 7x4 receptive field. Hence the PixelCNN decoder we used is both deeper and wider than that used in Chen et al. (2017a).

## 4.3 DENSITY ESTIMATION

Density estimation performance on CIFAR-10 of MAEs with different hyperparameters is provided in Table 3, compared with the top-performing likelihood-based unconditional generative models (first section) and variationally trained latent-variable models (second section). MAE models obtain improvement over the VLAE re-implemented by us, and slightly fall behind the original one in Chen et al. (2017a). Compared with PixelSNAIL (Chen et al., 2017b), the state-of-the-art auto-regressive generative model, the performance of MAE models is around 0.11 bits/dim worse. Further improving the density estimation performance of MAEs on natural images has been left to future work.

## 4.4 IMAGE RECONSTRUCTION AND GENERATION

We also investigate learning informative representations on CIFAR-10 dataset. The visualization of image reconstruction and generation is shown in Figure 4, together with VLAE for comparison. It is interesting to note that MAE tends to preserve rather detailed shape information than VLAE, whereas the color information, particularly the color for background, is partially omitted. One reasonable explanation, as discussed in Chen et al. (2017a), is that color is predictable locally. This serves as one

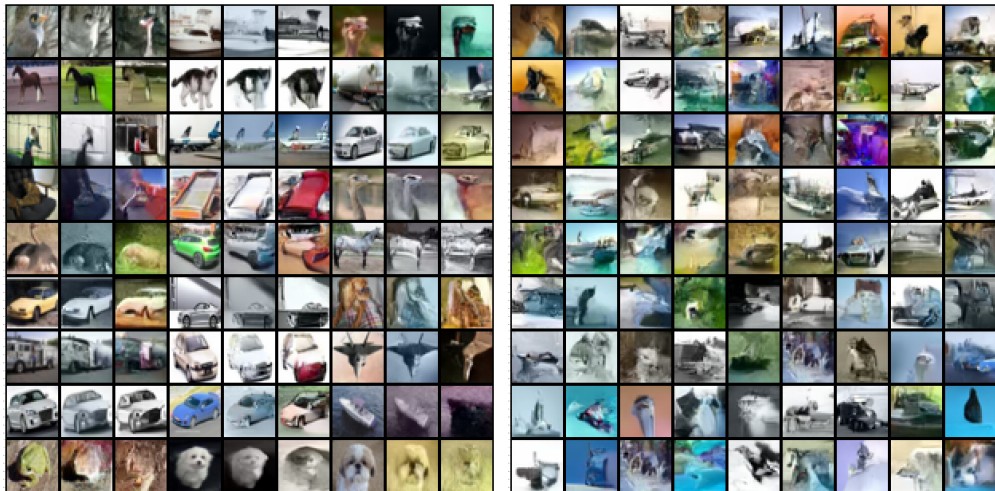

(a) CIFAR-10 reconstruction  (b) CIFAR-10 samples from MAE

Figure 4: Image reconstructions and samples on CIFAR-10. For reconstruction, original image is on the left, reconstructed image from MAE is in the middle, and VLAE is on the right.

example showing that MAEs can capture global structured information from data, omitting common correlations. Image samples from MAE are shown in Figure 4b.

## 5 CONCLUSION

In this paper, we proposed a mutual posterior-divergence regularization for VAEs, which controls the geometry of the latent space during training. By connecting the mutual posterior diversity with the mutual information, we have formally studied the theoretical properties of the proposed MAEs. Experiments on three benchmark datasets of images show the capability of MAEs on both density estimation and representation learning, with state-of-the-art or comparable likelihood, and superior performance on image reconstruction, unsupervised clustering and semi-supervised classification against previous top-performing models.

One potential direction for future work is to extend MAE to other forms of data, in particular text on which VAEs suffer a more serious KL-varnishing problem. Another exciting direction is to formally study the properties of the standard deviation of the mutual posterior KL-divergence used to measure smoothness, hence providing further justification of the proposed regularizer, or even introducing alternatives to further improve performances.

## ACKNOWLEDGEMENTS

The authors thank Zihang Dai, Junxian He, Di Wang and Zhengzhong Liu for their helpful discussions. This research was supported in part by DARPA grant FA8750-18-2-0018 funded under the AIDA program. Any opinions, findings, and conclusions or recommendations expressed in this material are those of the authors and do not necessarily reflect the views of DARPA.

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

APPENDIX: MAE: MUTUAL POSTERIOR-DIVERGENCE REGULARIZATION FOR VARIATIONAL AUTOENCODERS

## A  PROOF OF THEOREM 1

*Proof.*

$$MPD = \mathrm{E}_{X_1,X_2 \sim P(X)}[\mathrm{KL}(q_\phi(Z|X_1)||q_\phi(Z|X_2))]$$
$$= \mathrm{E}_{X_1,X_2 \sim P(X)}[\mathrm{H}(q_\phi(Z|X_1), q_\phi(Z|X_2)) - \mathrm{H}(q_\phi(Z|X_1))]$$

where $\mathrm{H}(\cdot)$ denotes the entropy. Then,

$$\mathrm{E}_{X_1,X_2 \sim P(X)}[\mathrm{H}(q_\phi(Z|X_1))] = \mathrm{E}_{P(X)}[\mathrm{H}(q_\phi(Z|X))]$$

and

$$\mathrm{E}_{X_1,X_2 \sim P(X)}[\mathrm{H}(q_\phi(Z|X_1), q_\phi(Z|X_2))]$$
$$= \mathrm{E}_{X_1,X_2 \sim P(X)}\left[-\int_{\mathcal{Z}} q_\phi(z|x_1) \log q_\phi(z|x_2)dz\right]$$
$$= \mathrm{E}_{P(X_2)}\left[-\iint p(x_1)q_\phi(z|x_1) \log q_\phi(z|x_2)dz \, dx_1\right]$$
$$= \mathrm{E}_{P(X_2)}\left[-\int \left(\int p(x_1)q_\phi(z|x_1)dx_1\right) \log q_\phi(z|x_2)dz\right]$$
$$= \mathrm{E}_{P(X_2)}\left[-\int q_\phi(z) \log q_\phi(z|x_2)dz\right]$$
$$= \mathrm{E}_{P(X)}[\mathrm{H}(q_\phi(Z), q_\phi(Z|X))]$$

So we have,

$$MPD = \mathrm{E}_{P(X)}[\mathrm{H}(q_\phi(Z), q_\phi(Z|X)) - \mathrm{H}(q_\phi(Z|X))]$$
$$= \mathrm{E}_{P(X)}[\mathrm{H}(q_\phi(Z), q_\phi(Z|X)) - \mathrm{H}(q_\phi(Z)) + \mathrm{H}(q_\phi(Z)) - \mathrm{H}(q_\phi(Z|X))]$$
$$= \mathrm{E}_{P(X)}[\mathrm{KL}(q_\phi(z|x)||q_\phi(z)) + \mathrm{KL}(q_\phi(z)||q_\phi(z|x))]$$

$\square$

## B  DETAILED DESCRIPTION OF EXPERIMENTS

### B.1  EXPERIMENTS FOR BINARY IMAGES

#### B.1.1  NEURAL NETWORK ARCHITECTURES AND TRAINING

The neural network architectures, including most of the hyperparameters, are the same as those in Chen et al. (2017a). The only difference in network architecture is the filter size of the PixelCNN decoder, which has been described in §4. For ResNet VAE with AF, we use the same ResNet encoder but a symmetric ResNet architecture for decoder. For encoder, we only use one stochastic layer with 32 dimensions.

In term of training, we use Adam optimizer (JLB, 2015) with learning rate 0.001, instead of Adamax used in Chen et al. (2017a). 0.01 nats/data-dim free bits was used in all the experiments. In order to get a relatively accurate approximation of $\mathcal{L}_{diverse}$ and $\mathcal{L}_{smooth}$, we used a much larger batch size 100 in our experiments. Polyak averaging (Polyak & Juditsky, 1992) was used to compute the final parameters, with $\alpha = 0.999$.

#### B.1.2  DETAILED RESULTS ON DENSITY ESTIMATION

Table 4 shows the detailed results of density estimation on MNIST. We see that increasing $\eta$ always achieving more informative latent $Z$, but the NLL not always becomes better. It illustrates the hypothesis that good representations should encode global structured information in the data, rather than local dependencies. It is interesting to see that, the effect of $\gamma$ on the latent $Z$ is inconsistent — increasing $\gamma$ from 0.1 to 0.5 leads to more informative $Z$ (larger KL) and better NLL, but too large $\gamma$

Table 4: Density estimation results on dynamically binarized MNIST. RE and KL correspond to the reconstruction error and the KL term in ELBO. MPD is the mutual posterior diversity in (4), and STD is the corresponding standard deviation in (6).

| Model | RE | KL | MPD | STD | $\mathcal{L}_{elbo}$ | NLL |
|---|---|---|---|---|---|---|
| ResNet VAE with AF | 56.04 | 25.38 | 1,193.18 | 630.90 | 81.42 | 79.28 |
| VLAE (w.o free bits) | 71.74 | 7.07 | 109.31 | 66.52 | 78.81 | 78.45 |
| VLAE (w. free bits) | 69.60 | 9.02 | 132.00 | 66.30 | 78.62 | 78.26 |
| MAE ($\eta = 0.5, \gamma = 0.1$) | 69.58 | 9.57 | 99.65 | 22.03 | 79.15 | 78.04 |
| MAE ($\eta = 1.0, \gamma = 0.1$) | 67.95 | 11.38 | 124.80 | 26.40 | 79.33 | 78.02 |
| MAE ($\eta = 2.0, \gamma = 0.1$) | 66.83 | 12.67 | 148.84 | 29.88 | 79.50 | 78.02 |
| MAE ($\eta = 0.5, \gamma = 0.5$) | 68.44 | 10.44 | 55.09 | 8.93 | 78.88 | 78.00 |
| MAE ($\eta = 1.0, \gamma = 0.5$) | 67.40 | 11.54 | 79.71 | 12.69 | 78.94 | 77.98 |
| MAE ($\eta = 2.0, \gamma = 0.5$) | 66.54 | 12.67 | 103.30 | 16.32 | 79.04 | 77.99 |
| MAE ($\eta = 0.5, \gamma = 1.0$) | 71.08 | 8.41 | 30.64 | 4.78 | 79.49 | 78.36 |
| MAE ($\eta = 1.0, \gamma = 1.0$) | 69.34 | 10.19 | 55.59 | 8.51 | 79.53 | 78.15 |
| MAE ($\eta = 2.0, \gamma = 1.0$) | 68.03 | 11.65 | 80.87 | 12.19 | 79.68 | 78.06 |

Table 5: Performance of semi-supervised classification using SVM classifiers with linear and RBF kernels. For each experiment, we report the average and standard deviation over 5 runs.

| Model | SVM-Linear | | | SVM-RBF | | |
|---|---|---|---|---|---|---|
| | 100 | 1000 | All | 100 | 1000 | All |
| ResNet VAE w. AF | 88.7±1.5 | 95.8±0.2 | 98.3±0.0 | 32.1±11.9 | 93.9±0.7 | 98.2±0.0 |
| VLAE | 84.9±1.8 | 94.3±0.2 | 96.7±0.0 | 51.5±5.8 | 90.9±0.4 | 97.0±0.0 |
| MAE: $\eta = 1.0, \gamma = 0.1$ | **91.9**±1.3 | 96.4±0.3 | 98.5±0.0 | 48.7±11.0 | 93.2±0.7 | 98.3±0.0 |
| MAE: $\eta = 0.5, \gamma = 0.5$ | 90.7±1.8 | 96.3±0.1 | 98.4±0.0 | 60.5±8.5 | 95.5±0.3 | 98.6±0.0 |
| MAE: $\eta = 1.0, \gamma = 0.5$ | 91.2±1.2 | **96.6**±0.2 | **98.5**±0.0 | **74.3**±5.8 | **96.5**±0.2 | **98.8**±0.0 |
| MAE: $\eta = 2.0, \gamma = 0.5$ | 90.3±1.5 | 96.5±0.1 | 98.5±0.0 | 51.5±8.9 | 92.3±1.4 | 98.5±0.0 |
| MAE: $\eta = 2.0, \gamma = 1.0$ | 90.3±1.5 | 96.5±0.1 | 98.5±0.0 | 54.5±9.8 | 95.0±0.5 | 98.5±0.0 |

(1.0) prevent the latent $Z$ to learn more information from data (smaller KL), resulting worse NLL. Hence, in practice considerations on the trade-off between diversity and smoothness of the latent space are needed.

### B.1.3 DETAILED RESULTS ON SEMI-SUPERVISED CLASSIFICATION WITH SVMS

Table 5 provides the performance of semi-supervised classification using SVMs with two different kernels — linear and RBF. MAE achieves the best classification accuracy on all the settings. It shoulld be noted that the accuracies of SVMs with non-linear kernels fluctuate more rapidly than linear ones, particularly when the number of labeled training data is small.

### B.1.4 LATENT SPACE VISUALIZATION

Figure 5 visualize the latent spaces of VAEs and MAEs with different settings on MNIST, by t-Distributed Stochastic Neighbor Embedding (t-SNE) (Maaten & Hinton, 2008). The first row displays the visualizations of ResNet VAE, VLAE without free-bits training and VLAE with free-bits training. The following three rows display visualizations of MAEs with $\gamma \in [0.1, 0.5, 1.0]$ and $\eta \in [0.5, 1.0, 2.0]$. We see that large $\eta$ encourages the posteriors to diverse from each other, by splitting the data into small groups. Meanwhile, increasing $\gamma$ is effective to prevent the phenomenon.

### B.2 EXPERIMENTS FOR CIFAR-10

Following Kingma et al. (2016) and Chen et al. (2017a), latent codes are represented by 16 feature maps of 8x8. Prior distribution is factorized Gaussian transformed by 8 auto-regressive flows, each of which is implemented by 3-layer masked CNNs (Oord et al., 2016) with 128 feature maps. Between every other auto-regressive flow, the ordering of stochastic units is reversed. PixelCNN++ (Salimans et al., 2017) with 7x3 receptive field is used as the decoder. Due to the limitation of computational

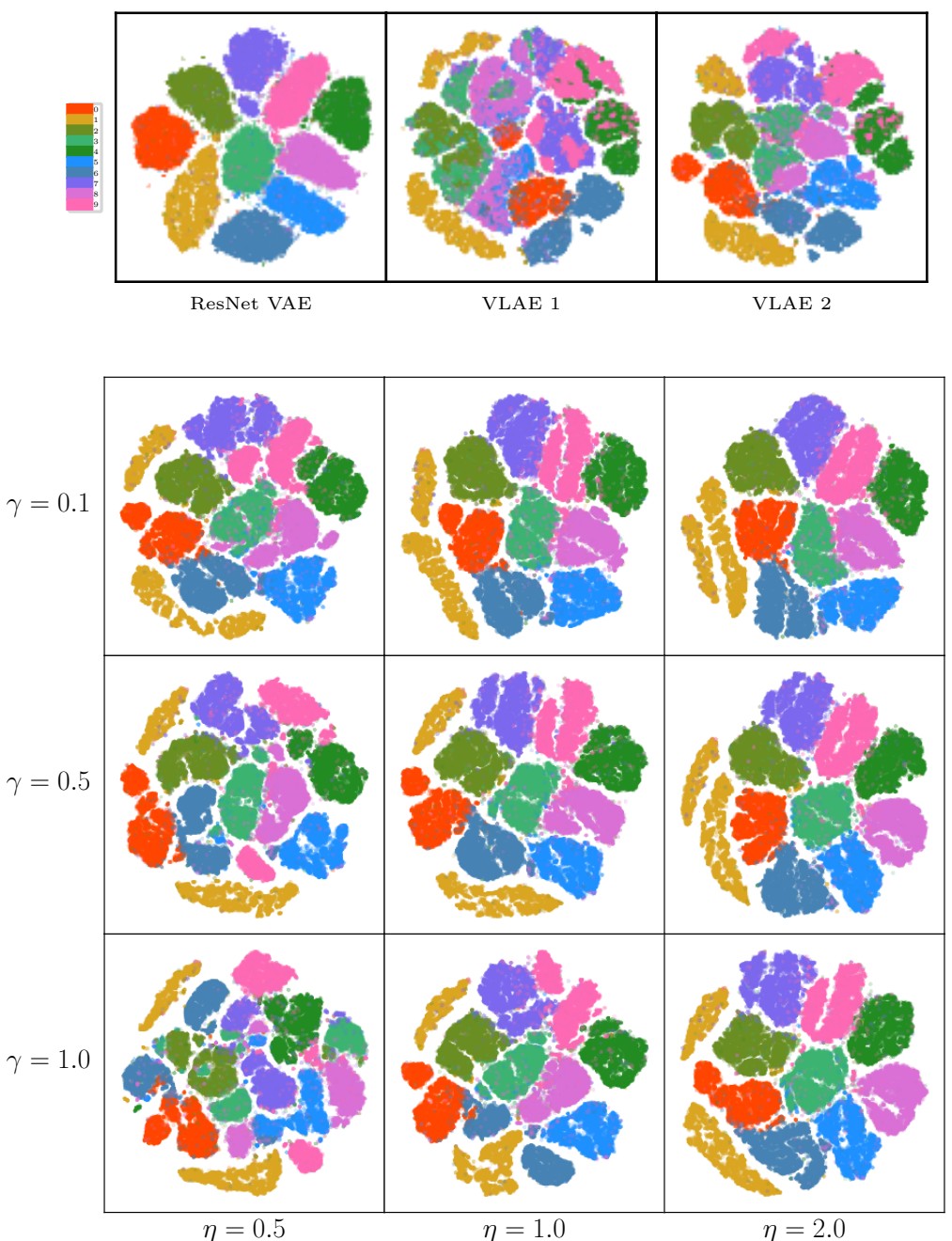

Figure 5: Visualization of latent space via t-SNE.

resources, we used batch size 64 in our experiments. Same as experiments on binary images, Polyak averaging (Polyak & Juditsky, 1992) was used to compute the final parameters, with $\alpha = 0.999$.

## C  GENERATED SAMPLES FROM VLAE AND MAE

Figure 6 provides generated images on MNIST, OMNIGLOT and CIFAR-10 from VLAE and MAE.

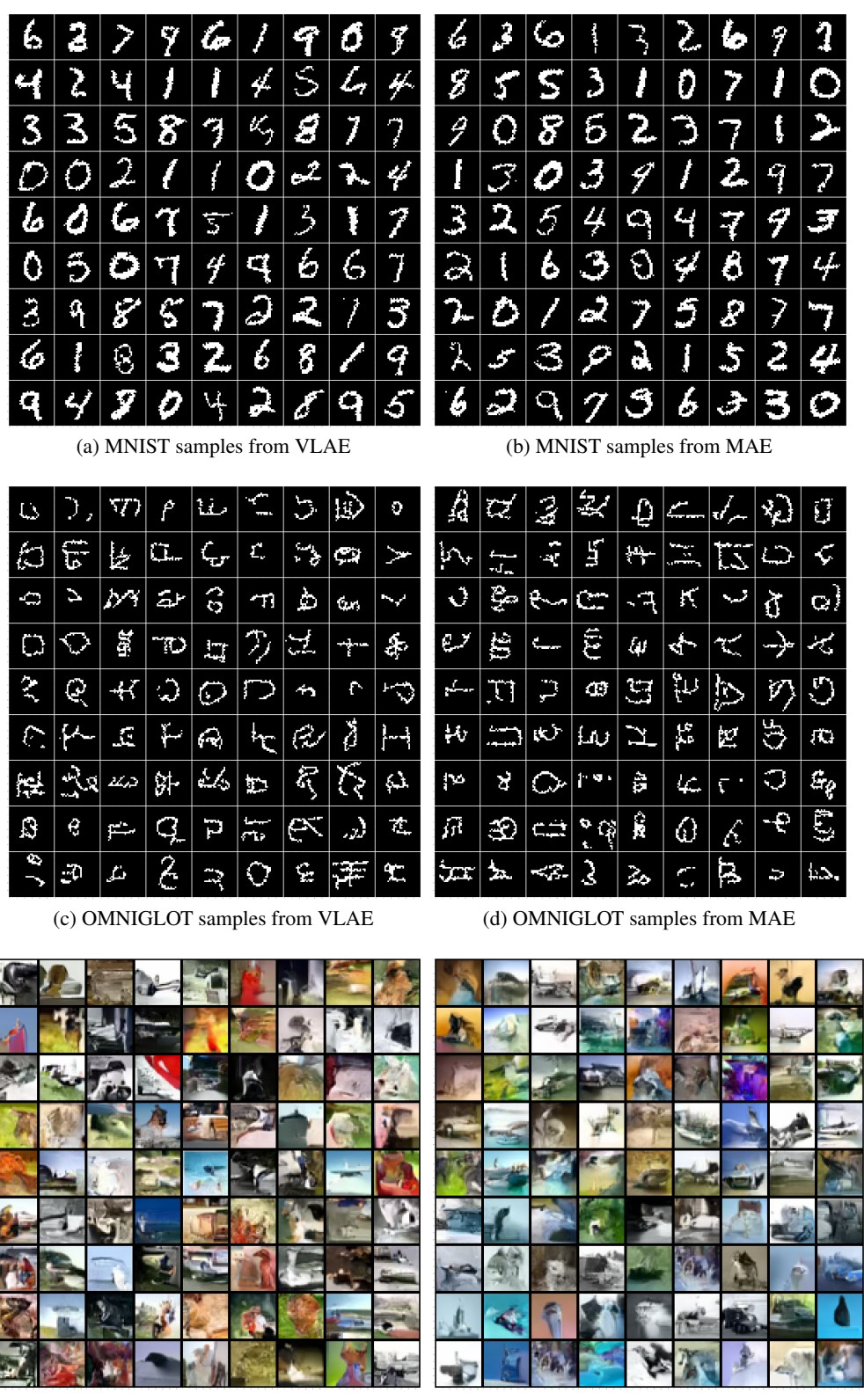

(a) MNIST samples from VLAE

(b) MNIST samples from MAE

(c) OMNIGLOT samples from VLAE

(d) OMNIGLOT samples from MAE

(e) CIFAR-10 samples from VLAE

(f) CIFAR-10 samples from MAE

Figure 6: Image samples from VLAE and MAE.

