# OpenReview forum: "MAE: Mutual Posterior-Divergence Regularization for Variational AutoEncoders"
_ICLR.cc/2019/Conference_

### Official Review · AnonReviewer2 · 2018-10-29
**Good paper, but the experiments could be improved**

**Rating:** 6
**Confidence:** 4

**Review:**

In this paper the authors present mutual posterior divergence regularization, a data-dependent regularization for the ELBO that enforces diversity and smoothness of the variational posteriors. The experiments show the effectiveness of the model for density estimation and representation learning.
This is an interesting paper dealing with the important issues of fully exploiting the stochastic part of VAE models and avoiding inactive latent units in the presence of very expressive decoders. The paper reads well and is well motivated.

The authors claim that their method is "encouraging the learned variational posteriors to be diverse". While it is important to have models that can use well the latent space, the constraints that are encoded seem too strong. If two data points are very similar, why should there be a term encouraging their posterior approximation to be different? In this case, their true posteriors will be in fact be similar, so it seems counter-intuitive to force their approximations to be different.

The numerical results seem promising, but I think they could be further improved and made more convincing.
- For the density estimation experiments, while there is an improvement in terms of NLL thanks to the new regularizer, it is not clear which is the additional computational burden. How much longer does it takes to train the model when computing all the regularization terms in the experiments with batch size 100?
- I am not completely convinced by the claims on the ability of the regularizer to improve the learned representations. K-means implicitly assumes that the data manifold is Euclidean. However, as shown for example by [Arvanitidis et al. Latent space oddity: on the curvature of deep generative models, ICLR 2018] and other authors, the latent manifold of VAEs is not Euclidean, and curved riemannian manifolds should be used when computing distances and performing clustering. Applying k-means in the high dimensional latent spaces of ResNet VAE and VLAE does not seem therefore a good idea.
One possible reason why your MAE model may perform better in the unsupervised clustering of table 2 is that the terms added to the elbo by the regularizer may force the space to be more Euclidean (e.g. the squared difference term in the Gaussian KL) and therefore more suitable for k-means.
- The semi-supervised classification experiment is definitely better to assess the representation learning capabilities, but KNN suffers with the same issues with the Euclidean distance as in the k-means experiments, and the linear classifier may not be flexible enough for non-euclidean and non-linear manifolds. Have you tried any other non-linear classifiers?
- Comparisons with other methods that aim at making the model learn better representation (such as the kl-annealing of the beta-vae) would be useful.
- The lack of improvements on the natural image task is a bit concerning for the generalizability of the results.

Typos and minor comments:
- devergence -> divergence in introduction
- assistant -> assistance in 2.3
- the items (1) and (2) in 3.1 are not very clear
- set -> sets in 3.2
- achieving -> achieve below theorem 1
- cluatering -> clustering in table 2

---

> ### Author Response · Authors · 2018-11-13
> **Response to Review 2**
>
> Thank you for the insightful comments!
> -- For your questions and concerns about the results on CIFAR-10, please see this post:
> https://openreview.net/forum?id=Hke4l2AcKQ&noteId=BylQ2fjL6X
> where we show stronger performance of our model.
>
> -- For your questions about the motivation of our method:
>  “encouraging the learned variational posteriors to be diverse” is the motivation of L_diversity. If we only have L_diversity in our regularization method, it is, as in your comment, counter-intuitive for similar data points. However, by adding the smoothness term L_smooth, we expect that the model itself is able to learn how to balance diversity and smoothness to capture both diverse patterns in different data points and shared patterns in similar ones. And our experimental results show that these two regularization terms together help achieve stronger performance.
>
> -- For your questions about additional computational burden:
> In order to train the model with large batch size, like 100, it requires more memory. But the computation of all the regularization terms is relatively efficient comparing to the computation of other parts of the objective. And the model converges as fast as that without the regularization.
>
> -- We really appreciate your comments about the evaluation of the learned representations.
> We agree that the latent manifold of VAEs may not be Euclidean.
> However, as discussed in our paper and previous works, good latent representations need to capture global structured information and disentangle the underlying causal factors, tease apart the underlying dependencies of the data, so that it becomes easier to understand, to classify, or to perform other tasks. Evaluating learned representations with unsupervised or semi-supervised methods with limited capacity is a reasonable way and has been widely adopted by previous works. From this perspective, it might be an important advantage of our method if our regularizer can force the space to be more Euclidean, because the learned representations are easier to be interpreted and utilized. Flexible classifiers might favor representations by just memorizing the data, thus not providing fair evaluation of the learned representations.

---

> > ### Comment · AnonReviewer2 · 2018-11-15
> > **Response to author feedback**
> >
> > Thank you for your clarifications and the additional experiments. As a result of these, I have increased my score by one point.
> >
> > I agree with your comments on the importance of learning interpretable and disentangled representation. However notice that this can also be achieved learning simple non-Euclidean spaces, that may require however a simple but non-linear classifiers (e.g. 1-layer neural network with a small number of hidden units, non-linear SVM).

---

> > > ### Author Response · Authors · 2018-11-16
> > > **Response to Review 2**
> > >
> > > Thank you for upgrading your score!
> > > We really appreciate your suggestion to evaluate learned representations with simple non-linear classifiers.
> > > We are performing experiments with SVM using non-linear kernels and will update results soon.

---

### Official Review · AnonReviewer3 · 2018-10-30
**Improving InfoVAE**

**Rating:** 6
**Confidence:** 4

**Review:**

This paper presents a new regularization technique for VAEs similar in motivation and form to the work on InfoVAE.  The basic intuition is to encourage different training samples to occupy different parts of z-space, by maximizing the expected KL divergence between pairwise posteriors, which they call Mutual Posterior-Divergence (MPD).  They show that this objective is a symmetric version (sum of the forward and reverse KL) of the Mutual Info regularization used by the InfoVAE.  In practice however, they do not actually use this objective.  They use a different regularization which is based on the MPD loss but they say is more stable because it's always greater than zero, and ensures that all latent dimensions are used.  In addition to the MPD based term, they also add another term which encouraging the pairwise KL-divergences to have a low standard-deviation, to encourge more even spreading over the z-space rather than the clumpy distribution that they observed with only the MPD based term.

They show state of the art results on MNIST and Omniglot, improving over the VLAE.  But on natural data (CIFAR10), their results are worse than VLAE.

Pros:
	1. The technique has a nice intuitive (but not particularly novel) motivation which is kinda-sorta theoretically motivated if you squint at it hard enough.
	2. The results on the simple datasets are solid and encouraging.

Cons:
	1.  The practical implementation is a bit ad-hoc and requires turn two additional hyper parameters (like most regularization techniques).
	2. The basic motivation and observations are the same as InfoVAE, so it's not completely novel.
	3. The CIFAR10 results are bit concerning, and one can't help but wondering if the technique really only helps when the data has simpler shared structure.

Overall:  I think the idea is interesting enough, and the results encouraging enough to be just above the bar for acceptance at ICLR.

I have the following question for the authors:

	1. Why do you use the truncated pixelcnn on CIFAR10?  Did you try it with the more expressive decoder (as was used on the binary images) and got worse results?  or is there some other justification for this difference?

I would have like to see the following modifications to the paper:

	1. The paper essentially presents two related but separate regularization techniques.  It would be nice to have ablation results to show how each of these perform on their own.
	2. Bonus points for showing results which combine VLAE (which already has a form of the MPD regularization) with the smoothness regularization.
	3. It would be nice to see samples from VLVAE in Figure 3 next to the MAE samples to more easily compare them directly.
	4. There are many grammatical and English mistakes.  The paper is still quite readably, but please make sure the paper is proofread by a native English speaker.

---

> ### Author Response · Authors · 2018-11-13
> **Response to Review 3**
>
> Thank you for the insightful comments!
>
> For your questions and concerns about the results on CIFAR-10 with more expressive decoders, please see this post:
> https://openreview.net/forum?id=Hke4l2AcKQ&noteId=BylQ2fjL6X
> where we show stronger performance with more expressive decoders for our model.
>
> For your specific questions,
> 1 & 2. We appreciate your suggestion to perform ablation experiments for the two terms in our regularizer. Actually, both of the regularization terms play important roles. Without L_smooth, the model will easily place different posteriors into isolated points far away from each other, obtaining L_diversity close to zero, and the model performance on both density estimation and representation learning is worse than original VLAE without the regularization. Moreover, removing the L_smooth term, the training of the model becomes unstable.
>
> 3. Thanks for your suggestion, we have added samples from VLAE in the updated version.
>
> 4. Thanks for your comment, we have revised the paper to fix the grammatical mistakes.

---

> > ### Comment · AnonReviewer3 · 2018-11-22
> > **Thanks for your response.**
> >
> > The changes to the paper look great, thanks for your updates.  They do not, however, change my basic opinion of the paper and so I will maintain my score as is.

---

### Official Review · AnonReviewer1 · 2018-11-03
**Interesting paper with marginal results**

**Rating:** 7
**Confidence:** 5

**Review:**

This paper proposes changes to the ELBO loss used to train VAEs, to avoid posterior collapse. They motivate their additional components rather differently than what has been done in the literature so far, which I found quite interesting.
They compare against appropriate baselines, on MNIST and OMNIGLOT, in a complete way.

Overall, I really enjoyed this paper, which proposed a novel way to regularise posteriors to force them to encode information. However, I have some reservations (see below), and looking squarely at the results, they do not seem to improve over existing models in a significant manner as of now.

Critics:
1.	The main idea of the paper, in introducing a measure of diversity, was well explained, and is well supported in its connection to the Mutual Information maximization framing. One relevant citation for that is Esmaeili et al. 2018, which breaks the ELBO into its components even further, and might help shed light on the exact components that this new paper are introducing. E.g. how would MAE fit in their Table A.2?
2.	On the contrary, the requirement to add a “Measure of Smoothness” was less clear and justified. Figure 1 was hard to understand (a better caption might help), and overall looking at the results, it is even unclear if having L_smooth is required at all?
 Its effect in Table 1, 2 and 3 look marginal at best?
Given that it is not theoretically supported at all, it may be interesting to understand why and when it really helps.
3.	One question that came up is “how much variance does the L_diverse term has”? If you’re using a single minibatch to get this MC estimate, I’m unsure how accurate it will be. Did changing M affect the results?
4.	L_diverse ends up being a symmetric version of the MI. What would happen if that was a Jensen-Shannon Divergence instead? This would be a more principled way to symmetrically compare q(z|x) and q(z).
5.	One aspect that was quite lacking from the paper is an actual exploration of the latent space obtained.  The authors claim that their losses would control the geometry of the latents and provide smooth, diverse and well-behaved representations. Is it the case?
 Can you perform latent traversals, or look at what information is represented by different latents?
This could actually lend support to using both new terms in your loss.
6.	Reconstructions on MNIST by VLAE seem rather worst than what can be seen in the original publication of Chen et al. 2017? Considering that the re-implementation seems just as good in Table 1 and 3, is this discrepancy surprising?
7.	Figure 2 would be easier to read by moving the columns apart (i.e. 3 blocks of 3 columns).

Overall, I think this is an interesting paper which deserves to be shown at ICLR, but I would like to understand if L_smooth is really needed, and why results are not much better than VLAE.

Typos:
-	KL Varnishing -> vanishing surely?
-	Devergence -> divergence

---

> ### Author Response · Authors · 2018-11-13
> **Response to Review 1**
>
> Thank you for the insightful comments!
> For your questions:
> 1. Thanks for pointing out the related work. We cited  Esmaeili’s paper in our updated version. Actually, MAE does not fit anyone in their Table A.2. If we also decompose our objective in the same, our objective is, if we use the original form of MPD and ignore L_sommth, term (1) + (2) + (4’), where (4’) is a modified version of (4).
> The original (4) is KL(q(z) || p(z)) = E_q(z} [log q(z) - log p(z)], while (4’) is E_{p(x) q(z)} [log q(z|x) - log p(z)]
>
> 2. In our experiments, L_smooth plays a very important role. If we remove it, the model will easily place different posteriors into isolated points far away from each other, obtaining L_diversity close to zero. This phenomenon becomes more serious when a more powerful prior is applied, like auto-regressive flow. The unsupervised clustering and semi-supervised classification experiments justified the necessity of L_smooth. We also visualized the latent spaces with different settings in Appendix B.1.3, which might be helpful to understand the effects of the two regularization terms.
>
> From the theoretical perspective, we have not provided rigorous support of L_smooth and will leave it to future work.
>
> 3. In order to better approximate L_diversity, we used large batch size in our experiments. For binary images, we use batch size 100. For natural images, due to memory limits, we use 64. The details are provided in Appendix. In practice, we found that these batch sizes provide stable estimation of L_diversity.
>
> 4. As we discussed in the paper, one advantage of our regularization method is that L_diversity is computationally efficient. Previous works such as InfoVAE and AAE also has considered the  Jensen-Shannon Divergence. But directly optimizing it is intractable, and they applied adversarial learning methods.
>
> 5. We plan to show the reconstruction results with linearly interpolated z-vectors in another updated version. We appreciate your suggestions if there are better ways of investigating the latent space in terms of "latent travelsals".
>
> 6. The possible reason that VLAE obtained worse reconstruction than the original paper is that in our experiments, we used more powerful decoders with more layers and receptive fields. We want to test the performance of our regularizer with sufficiently expressive decoders. With more powerful decoders, our reimplementation of VLAE achieved better NLL but worse reconstruction, showing that VLAE suffers the KL varnishing issue with stronger decoders.
>
> 7. Thanks for your suggestion! We will make figure 2 easier to understand and update the revised version later.

---

> > ### Comment · AnonReviewer1 · 2018-11-24
> > **Results no longer look marginal, thanks for the extra work!**
> >
> > Thanks to the authors for the work in addressing my questions and comments.
> >
> > 2. That’s interesting to know, makes sense indeed. I would explicitly indicate this in your “Measure of Smoothness.” section then, as this does not come across in the current text.
> > The new figure in Appendix B.1.3 is interesting to see, but does not seem to indicate such a drastic effect, which I guess might be due to t-SNE “fixing it”, but I am not sure what would be the best way to showcase this effect.
> >
> > 5. Yes sorry what I meant by “latent traversals” is something akin to the single unit clamping done in Beta-VAE (Higgins et al 2017, https://openreview.net/forum?id=Sy2fzU9gl). In your case, given you have latents with 32 dimensions this is harder to do easily, hence interpolations might be interesting to see indeed.
> >
> > I think the updated results seem to make the model stronger and show visible improvements on VLAE.
> > I am still a bit unclear on the exact characteristics of the latent space learnt and I’m looking forward to see more work in that direction.
> >
> > Hence the paper does seem good enough in its current state, so I’d recommend publication as a poster (keeping my score, increasing my confidence).

---

### Public Comment · ~Hello_Kitty2 · 2018-10-02
**suggestion**

 This work is closely related to the following work:
 R. D. Hjelm et al, "Learning deep representations by mutual information estimation and maximization", https://arxiv.org/abs/1808.06670.

  I would suggest the authors cite the latest work and compare the performance between the two methods.

  By the way, in the appendix, it mentions that the KL divergence is equal to H(.,.) - H(.), where H(.,.) denotes the relative entropy. Note that relative entropy is actually the KL divergence. Please use a proper name to define H(.,.). The different information measures can be found in
  1. Cover and Thomas, "Elements of Information Theory".
  2. Raymond Yeung, "Information Theory and Network Coding"
  3. Robert Gallager, "Information Theory and Reliable Communication"

---

> ### Author Response · Authors · 2018-10-02
> **thanks for pointing out missing related work**
>
> Thanks for pointing out the related work missed in the paper.
> We will cite and compare with it in our revised version.
>
> We really appreciate your comments about the notation used in the appendix.
> We will revise it.

---

### Author Response · Authors · 2019-01-06
**Carema-Ready version updated**

Carema-Ready version of this paper has been uploaed

---

### Meta-Review · Area_Chair1 · 2018-12-14

**Confidence:** 4
**Recommendation:** Accept (Poster)

**Metareview:**

This paper proposes a solution for the well-known problem of posterior collapse in VAEs: a phenomenon where the posteriors fail to diverge from the prior, which tends to happen in situations where the decoder is overly flexible.

A downside of the proposed method is the introduction of hyper-parameters controlling the degree of regularization. The empirical results show improvements on various baselines.

The paper proposes the addition of a regularization term that penalizes pairwise similarity of posteriors in latent space. The reviewers agree that the paper is clearly written and that the method is reasonably motivated. The experiments are also sufficiently convincing.